# Familial Adult Myoclonus Epilepsy: A Non-Coding Repeat Expansion Disorder of Cerebellar–Thalamic–Cortical Loop

**DOI:** 10.3390/cells12121617

**Published:** 2023-06-13

**Authors:** Claudia Cuccurullo, Pasquale Striano, Antonietta Coppola

**Affiliations:** 1Department of Neuroscience, Reproductive Sciences and Odontostomatology, Federico II University of Naples, 80131 Naples, Italy; antonietta.coppola@unina.it; 2Pediatric Neurology and Muscular Diseases Unit, IRCCS Istituto Giannina Gaslini, 16147 Genova, Italy; 3Department of Neurosciences, Rehabilitation, Ophthalmology, Genetics, Maternal and Child Health, University of Genova, 16126 Genova, Italy

**Keywords:** cortical tremor, myoclonus, epilepsy, FAME, TTTCA/ATTTC intronic repeats, RNA foci

## Abstract

Familial adult myoclonus Epilepsy (FAME) is a non-coding repeat expansion disorder that has been reported under different acronyms and initially linked to four main loci: FAME1 (8q23.3–q24.1), FAME 2 (2p11.1–q12.1), FAME3 (5p15.31–p15.1), and FAME4 (3q26.32–3q28). To date, it is known that the genetic mechanism underlying FAME consists of the expansion of similar non-coding pentanucleotide repeats, TTTCA and TTTTA, in different genes. FAME is characterized by cortical tremor and myoclonus usually manifesting within the second decade of life, and infrequent seizures by the third or fourth decade. Cortical tremor is the core feature of FAME and is considered part of a spectrum of cortical myoclonus. Neurophysiological investigations as jerk-locked back averaging (JLBA) and corticomuscular coherence analysis, giant somatosensory evoked potentials (SEPs), and the presence of long-latency reflex I (or C reflex) at rest support cortical tremor as the result of the sensorimotor cortex hyperexcitability. Furthermore, the application of transcranial magnetic stimulation (TMS) protocols in FAME patients has recently shown that inhibitory circuits are also altered within the primary somatosensory cortex and the concomitant involvement of subcortical networks. Moreover, neuroimaging studies and postmortem autoptic studies indicate cerebellar alterations and abnormal functional connectivity between the cerebellum and cerebrum in FAME. Accordingly, the pathophysiological mechanism underlying FAME has been hypothesized to reside in decreased sensorimotor cortical inhibition through dysfunction of the cerebellar–thalamic–cortical loop, secondary to primary cerebellar pathology. In this context, the non-coding pentameric expansions have been proposed to cause cerebellar damage through an RNA-mediated toxicity mechanism. The elucidation of the underlying pathological mechanisms of FAME paves the way to novel therapeutic possibilities, such as RNA-targeting treatments, possibly applicable to other neurodegenerative non-coding disorders.

## 1. Introduction

Familial adult myoclonus epilepsy is a rare autosomal dominant disorder featuring cortical hand tremors, myoclonic jerks, and, more rarely, convulsive seizures. Since its first description in 1985, FAME has been reported in over a hundred pedigrees worldwide, allowing the characterization of well-known clinical features [1]. However, FAME is still an underrecognized disorder and it is often misdiagnosed and mistreated.

The genetic defect consists of non-coding pentameric repeat expansions in distinct genes for nearly all cases of FAME worldwide. These expanded pentamers, independently from the affected gene, have been hypothesized to play a pathogenic role through an RNA-mediated mechanism determining dysfunction of the cerebellar–cerebral networks and specifically of the cerebellar–thalamic–cortical loop.

This hypothesis has also strengthened the previous evidence provided by the neuropathological and neurophysiological investigations suggesting that the cerebellum holds a central role underlying cortical hyperexcitability in FAME.

We review the clinical phenotype, treatment, and genetic characteristics of FAME and discuss the recent advances in understanding the pathogenesis of this condition as a cortical–subcortical network disorder.

### 1.1. History of the Disease across Different Acronyms

The term “cortical tremor” was introduced by Ikeda et al. [1] in 1990 to describe a fine jerky movement of hand extremities attaining pseudo-rhythmicity on posturing and, thus, resembling essential tremor, in two unrelated patients. They further presented with adult-onset generalized epilepsy and the absence of cerebellar or pyramidal signs. Considering its clinical and electrophysiological features (see Section 2), this involuntary movement was defined as a variant of cortical reflex myoclonus [2,3]. A hereditary disorder characterized by adult-onset cortical tremor, associated with myoclonic jerks and seizures, was described in Japanese families even before the introduction of the term “cortical tremor”.

### 1.2. Japanese Families

The first description of a family presenting with adult-onset fine finger tremulous movement, myoclonic jerks, and generalized seizures was reported by Uyama et al. [4] in 1985. This disorder affected the family therein described through three generations with high penetrance, suggesting a genetic etiology with an autosomal dominant mode of inheritance. Furthermore, the electrophysiological assessment demonstrated the presence of cortical reflex myoclonus, whilst no other neurological or neuroradiological features were noticed.

The same authors reported four additional unrelated families, including 27 affected members through three generations, with similar signs and symptoms and proposed the term FAME [5].

Forty patients from 12 families presenting with the same clinical characteristics were reported using the term “familial essential myoclonus and epilepsy” (FEME), although the neurophysiological studies were not performed [6]. Yasuda et al. reported two similar families with 26 affected members who showed autosomal dominant hand tremors, myoclonus, and seizures. Since this condition showed a non-progressive course, the authors used the term “benign adult-onset familial myoclonic epilepsy” (BAFME) [7].

Okuma et al. reported four additional families and proposed that the previous reports on similar pedigrees, although using different terms, described the same disorder. Since the involuntary movements of the patients clinically resembled a tremor rather than myoclonus, they suggested that the term “familial cortical tremor with epilepsy” (FCTE) would have been more appropriate [8,9]. Later, Terada et al. investigated six patients from three pedigrees with cortical tremors and proposed the term “familial cortical myoclonic tremor” (FCMT) rather than using those containing “epilepsy”, claiming that the patients only presented with occasional seizures [10].

In 1999 Mikami et al. and Plaster et al. identified the FAME locus on chromosome 8q23.3–q24.1 in five Japanese families and hypothesized the existence of a founder effect [11,12]. Since the growing knowledge at that time on the role of genes encoding ion channels in the etiology of several epilepsies, the authors suggested that these genes might have been responsible for FAME (see Section 6).

### 1.3. Non-Japanese Families

Among non-Japanese families, a similar clinical picture was initially described in European pedigrees and particularly in those of Italian origin. The first report was provided by Elia et al. [13], who reported a family presenting with cortical tremor, showing the neurophysiological features of cortical reflex myoclonus, and infrequent generalized seizures through five consecutive generations.

An Italian pedigree showing additional features to the core clinical and neurophysiological characteristics of FAME was described by Guerrini et al. This family was distinguished from the others by focal seizures and frontotemporal EEG abnormalities in a few family members. Genome-wide linkage analysis mapped this condition on chromosome 2p11.1–q12.2 [14]. The authors used the acronym ADCME (autosomal dominant cortical myoclonus and epilepsy) and proposed this condition as a new epilepsy syndrome, distinguished from that one described in Japanese families.

Nevertheless, three families from the South of Italy sharing the typical clinical and neurophysiological features of Japanese families were subsequently reported with linkage to chromosome 2p11.1–q12.2 as the ADCME pedigree [15,16]. Even though these families did not show the distinguishing features of ADCME, considering the overlapping syndromic core, the authors suggested that ADCME could be included within the same spectrum.

About other European families, a large Dutch pedigree with cortical tremor and epilepsy, in addition to intellectual disability in a more progressed stage [17], and a four-generation Spanish kindred were reported [18]. In both pedigrees, linkage to chromosome 8q23.3–q24.1 was excluded, and the Dutch one also linked to chromosome 2p11.1–q12.2 was excluded afterwards [19], further supporting the genetic heterogeneity of FAME.

In 2005, Striano P et al. [20] and van Rootselaar et al. [21] reviewed the fifty Japanese and European families with autosomal dominant cortical tremor, myoclonus, and epilepsy reported until then under different terms. They both concluded that FAME, FEME, BAFME, FCTE, FCMT, and ADCME represented the same clinical entity, even if genetically heterogeneous, with Japanese families linked to 8q24 and Italian ones to 2p11.1–q12. A third locus was later identified in a large French family, mapping to chromosome 5p15.31–p15.1 [22,23]. FAME pedigrees have also been subsequently described in China [24], even showing linkage to the French locus 5p15.31–p15.1 [25,26]. A fourth locus was lastly identified in a Thai family on chromosome 3q26.32–3q28 [27]. Further pedigrees worldwide have been reported in recent years, including a large family of European descent living in New Zealand and Australia [28], a family from Turkey [29], pedigrees from a South Indian Community [30,31], Sri Lanka [32] and a family of South African origin [33].

## 2. Clinical Features

The core features defining FAME include cortical tremors, epileptic myoclonus, and seizures. Despite variable degrees in the age of onset and severity, FAME usually manifests with cortical tremor and myoclonus within the second decade of life (range 11–50 years old) [34,35,36]. Epilepsy usually presents with bilateral tonic–clonic seizures by the third or fourth decade (range 12 and 67 years old). Seizures are rare and reported as the presenting symptom in 16% of the described patients [21,36].

### 2.1. Cortical Tremor and Myoclonus

Cortical tremor is the core feature of FAME, and consists of continuous, arrhythmic (or pseudo-rhythmic), fine myoclonic jerks, involving the distal upper limbs, bilaterally. It is induced by posture or action, although may be noticed even at rest [21].

Cortical tremor is stimulus sensitive [21] to tactile and occasionally photic stimulation [37,38]; it is enhanced by emotional stress, fatigue, or sleep deprivation [20,28,37], whilst alcohol responsiveness has been reported in a minority of cases [28,37,39]. Sensitivity to vibration and glucose deprivation has also been observed [39].

Severity varies between and within pedigrees and may interfere with daily life activities. There is no significant progression over time, despite a worsening of the disturbance that may be observed in the elderly (Figure 1) (see Section 3) [20,21].

The semiology of cortical tremor strictly resembles essential tremor, from which it is clinically distinguished for the lack of true rhythmicity and the irregular spatial distribution [20]. Furthermore, cortical tremor does not usually respond to beta-blockers, whilst it benefits from antiseizure medications (ASMs) [20,21]. Clinical criteria to distinguish between FAME and essential tremor have been suggested [39]. However, the differential diagnosis of these disorders requires the neurophysiological demonstration of the cortical origin of the tremor [20]. Patients usually present with further multifocal myoclonic jerks involving the proximal arms, axial and facial sites, particularly the eyelids, and the lower limbs [14,21,28,35,37,41]. They are equally enhanced by posture and action and evoked by sensory stimuli. As cortical tremors, these myoclonic jerks are generated by networks involving the sensory–motor cortex. Latorre et al. proposed that cortical tremor can be defined as a rhythmic variant of cortical myoclonus and that it belongs to a spectrum of cortical myoclonus, from reflex myoclonus to myoclonic epilepsy, resulting from sensorimotor cortex hyperexcitability [42].

### 2.2. Seizures

Seizures are infrequent (from 5 to 10 episodes during the entire lifespan) [35] and, when present, appear after the onset of cortical tremor and myoclonus. They usually consist of bilateral tonic–clonic seizures responsive to monotherapy, sometimes preceded by clusters of myoclonic jerks. Seizures may be triggered by sleep deprivation, emotional stress, and photic stimulation [5,15,20,28,38]. Focal onset seizures with impaired awareness and that are drug-resistant have also been reported in some pedigrees [14,22,31]. Even myoclonic seizures provoked by photic stimulation and causing falls have been described [18,28,43]. Absences and febrile seizures have been reported in a minority of cases [13,29,43].

### 2.3. Additional Clinical Features

Migraine has been reported in some pedigrees [22,28,29,44] and night blindness with a reduced b-wave response on electroretinography (ERG) has been described in a single pedigree from Japan [45]. However, migraine was not always associated with cortical tremor, as it is a frequent disorder that may coexist with FAME [39].

When involving the trunk and the lower limb, the myoclonus impairs autonomous walking in individuals from different families [23,39,46]. However, additional subtle cerebellar signs have been reported in some pedigrees, including postural ataxia, downbeat nystagmus, and impaired smooth pursuit, and have been hypothesized as part of the syndrome [22,47,48] (see Section 7).

Cognitive impairment has been reported in various pedigrees, especially in older members [13,14,18,28,41]. In detail, neuropsychological testing disclosed visuospatial impairment in some pedigrees, along with verbal and visual memory deficits in some individuals, indicative of temporal lobe dysfunction [41,46].

Furthermore, a relatively high frequency of psychiatric features in patients with FAME was observed [46,49]. These manifestations included generalized anxiety disorder and depression, along with pathological traits of personality (i.e., paranoid, psychasthenia, schizophrenia, hypomania) and impaired quality of life of these patients. Additionally, the myoclonus severity significantly correlated with both state and trait anxiety, while it did not correlate with depression symptoms [49]. Psychiatric comorbidity was described afterwards in non-Italian pedigrees, including a Turkish pedigree [29] and a South Indian pedigree [31].

## 3. Disease Course

Although FAME has been considered to have a benign course, unlike progressive myoclonus epilepsies (PMEs), the natural history of the disease includes a worsening of cortical tremor and myoclonus, along with ataxia and slow-progressive dementia [35,46,50]. Coppola et al. demonstrated that, although very slowly, the disease progresses over the years. The authors observed a slow but gradual and progressive worsening of the myoclonus, which was correlated with the duration of the disease. Myoclonus usually interfered with fine movements of daily life (i.e., handwriting, signing, or speech) in adult age, despite ASM treatment, and resulted in walking impairment for individuals aged > 80 years. However, none of the patients were wheelchair-bound or bedridden [46]. In Japanese patients, a progressive increase in cortical hyperexcitability revealed by increased SEP amplitudes was found to underlie the worsening of cortical tremor with age [50].

Furthermore, the long-term evolution of EEG revealed mild progressive slowing of the background activity in parallel with the gradual worsening of myoclonus over the years [46]. This finding was described afterwards also in Japanese patients, suggesting a mild diffuse brain dysfunction with minimal progression [51] (see Section 5).

## 4. Neurophysiological Investigations

Neurophysiological Investigations are essential to identify cortical tremors and for the diagnosis of FAME.

The clinical and electrophysiological features of this involuntary movement, as described by Ikeda et al. [1] include (1) a brief EMG burst of about 50 ms in duration; (2) distal predominance, mainly involving the fingers; (3) an increase by outstretched posture and particularly by action; (4) relative rhythmicity; and (5) no definite synchronization or reciprocity in the two antagonist muscles. Furthermore, this type of tremor fulfils the criteria of cortical reflex myoclonus [2,3]: (1) enlarged cortical components of somatosensory evoked potentials, named giant SEPs [3,52]; (2) enhancement of the long latency reflexes (LLRs) [53]; and (3) the presence of cortical spikes preceding EMG myoclonic jerks at the jerk-locked averaging (J-LBA) analysis [54].

### 4.1. EEG-EMG Polygraphy

A polygraphic recording is the first step to detecting tremor and cortical myoclonus since it allows us to analyze the relationship between myoclonic and cortical events (Figure 2) [54,55,56].

The EEG background activity is usually normal, between 11 and 12 Hz before age 40 years. However, a gradual and progressive slowing is evident over the years, independent from treatment [46]. Interictal EEG may show generalized epileptiform discharges (EDs) (i.e., generalized spike/polyspikes–waves) in a variable percentage of affected individuals, especially in those not receiving treatment [21,35,38,43,46], and photoparoxysmal response [38,43,46,57]. A faster frequency of generalized spike–waves in FAME (4.3 ± 1.0 Hz) has been hypothesized to be a distinctive EEG feature, compared with that in epilepsy with generalized tonic–clonic seizures only [57]. Focal EDs have been reported in a minority of cases, particularly in those presenting with drug-resistant focal seizures and initially termed ADCME [14]. EEG EDs and photosensitivity are usually evident during the intermediate phases of the disease, and they tend to disappear along the disease progression [46]. Furthermore, a photomyogenic response (i.e., muscular, mainly anterior, response synchronous with photic stimulation) may also be present [20]. A retrospective EEG and polysomnography analysis have shown a reduction of EDs during all sleep stages in FAME [58].

Surface EMG in patients with FAME is characterized by arrhythmic or semi-rhythmic and high-frequency (around 10/s) myoclonic jerks. The myoclonic bursts last about 50 ms, supporting a cortical origin [1,55,56]. Additionally, the jerks are synchronous between agonist and antagonist muscles, whilst in essential tremors, they show a rhythmic agonist/antagonist alternate [20]. Furthermore, EMG bursts show stimulus sensitivity consistent with cortical reflex myoclonus, since they are triggered by tendon tap, posturing, passive or voluntary movement, and emotional stress [35,55]. Cortical tremor induced by fixation-off sensitivity and exacerbated during drowsiness was described in a single patient [59].

### 4.2. Jerk-Locked Back Averaging (J-LBA) Analysis

The cortical event preceding the myoclonic jerk may be seen in raw recordings, usually as spikes or polyspikes. However, the cortical correlate of myoclonus may be merged with the background activity and, therefore, is not visible in raw recordings. In these conditions, the J-LBA analysis demonstrates that the myoclonus is of cortical origin. This neurophysiological technique commonly shows a positive–negative, biphasic spike or wave, on the contralateral sensorimotor regions, time-locked to the myoclonic events. The cortical potential usually precedes the myoclonic jerks with a latency of around 20 ms for the hand myoclonus and 30 ms for the leg myoclonus [20,55,56].

### 4.3. Cortico-Muscular Coherence Analysis

EEG-EMG corticomuscular coherence analysis may help demonstrate a cortical drive when the J-LBA analysis has not shown a clear-cut cortical event, particularly in patients with high-frequency cortical myoclonus [55,60,61]. It may also help differentiate cortical from essential tremor [62]. This analysis aims to measure the strength of functional coupling between the sensorimotor cortex activity and EMG signals recorded from contralateral upper limb muscles. In cortical myoclonus it commonly discloses an increased coherence between EMG and contralateral EEG fronto-central electrodes signals, especially in the beta band (peak at 17 Hz, range 8–30 Hz); phase spectra analysis confirms that EEG precedes EMG activity [55,61,62].

### 4.4. Giant Somatosensory-Evoked Potential (SEP) and High-Frequency Oscillations (HFOs)

Cortical tremor is considered a rhythmic variant of cortical reflex myoclonus and is characterized by abnormally enlarged SEP, in response to peripheral electric stimulation, namely, giant SEPs [1]. The giant SEP was initially described in PMEs and defined, when evoked at the median nerve, by an amplitude difference from the N20 peak to the P25 peak larger than 8.6 uV, or that measured from the P25 to the N33 peak larger than 8.4 uV [63]. In contrast, the initial negative component, N20, was found not enhanced, suggesting that the sensory input into the sensorimotor cortex and the primary cortical response were not altered [3,56,63].

Since its first description, the giant SEP was regarded as a manifestation of a pathologically enhanced excitability of those areas of the sensorimotor cortex, which generates normal SEPs [3]. The presence of the giant SEP, therefore, supports the diagnosis of cortical or cortical reflex myoclonus, as it is not seen in myoclonus of another origin [55]. However, its absence does not exclude the presence of cortical myoclonus. Indeed, the sensorimotor cortex hyperexcitability may be attenuated by ASMs (i.e., clonazepam, levetiracetam, perampanel, and valproate) [52,64,65]. In this case, the recovery cycle of SEPs (SEP-R) using paired-pulse stimuli may provide additional information [56].

Dubbioso et al. have observed a significant reduction of N20-P25 suppression at both cortical short intervals and subcortical longer interstimulus intervals (ISIs) in FAME2 patients. Furthermore, analysis of high-frequency oscillations (HFOs) on the N20 component revealed a smaller e-HFO area and a larger l-HFO area, compared to patients with JME and healthy controls. These findings supported the alteration of inhibitory circuits within the primary somatosensory cortex (S1) and the concomitant involvement of subcortical networks (see Section 7) [66]. Lastly, high-frequency oscillations (HFOs) on the P25 component, which are not physiological features unlike HFOs within N20, have been found in patients with FAME. They have been proposed as a biomarker for FAME diagnosis and may reflect the degree of cortical hyperexcitability [67].

### 4.5. Long-Latency Reflexes (LLRs)

LLRs are motor responses to somesthetic stimuli which follow transcortical pathways. After electrical stimulation of the hand (thenar eminence), three components can be identified: they are termed LLR I–II–III. LLR I is also called C-reflex. LLRs may be physiologically present during tonic muscle activation, after various stimuli. The presence of LLR I (or C reflex) at rest, usually with a latency of 40 ms, suggests hyperexcitability of the central part of the loop reflex, that is, the sensorimotor cortex. Therefore, the finding of the C reflex at rest is pathognomonic of reflex myoclonus, albeit not of its cortical origin [53,56], and can be considered as a marker of abnormal interaction between primary sensory cortex S1 and primary motor cortex M1 [42]. In FAME patients the C reflex at rest is commonly observed [20].

### 4.6. Transcranial Magnetic Stimulation (TMS)

In Guerrini et al., the resting motor threshold intensity was significantly reduced, and the post-motor evoked potential silent period was shortened. These findings suggested cortical hyperexcitability with impairment in inhibitory mechanisms within the primary motor cortex (M1) [14]. Subsequently, in a Dutch pedigree short-interval intracortical inhibition (SICI) was found significantly reduced, compatible with intracortical GABA_A_-ergic dysfunction. The decreased cortical inhibition was hypothesized to be caused by a dysfunction of the cerebellar input to inhibitory intracortical circuits via the cerebellar–thalamic–cortical loop [47]. These findings were confirmed in a further Italian pedigree [41]. Dubbioso et al. recently studied the cortical sensorimotor, and the thalamo–cortical networks in a large series of genetically confirmed FAME2 patients. The authors demonstrated the impairment of both GABA-A and GABA-B circuits, along with a marked increase in facilitatory glutamatergic circuits within M1. The inhibitory circuits were also altered within S1 and the concomitant involvement of subcortical networks was hypothesized to be involved in the pathophysiology of FAME (see Section 7). This study concluded that TMS measures, especially SICI and long interval intracortical inhibition, were demonstrated superior to SEP parameters for FAME diagnosis [66].

## 5. Neuroradiology

Brain MRI is usually normal, albeit structural brain abnormalities have been reported in a minority of cases. These include mild enlargement of the subarachnoid spaces or the lateral ventricles and slight cerebellar atrophy [13,18,41]. Two proton magnetic resonance (MR) spectroscopy (^1^H-MRS) studies disclosed an abnormal increase in the Cho/Cr ratio [68] and a significant decrease in NAA/Cho ratio [69] in FAME patients’ cerebellum, which reflects cerebellar neuron loss and secondary astrocytosis. On the contrary, there was no difference in the metabolite ratios, neither in the frontal and occipital cortex [68] nor in the thalamus [69]. These led the authors to hypothesize that the cerebellum could be the prominent site of neuronal dysfunction in FAME (see Section 7). Additionally, ^1^H-MRS was also able to detect brain changes in patients with recent disease onset and may be useful in the early diagnosis of FAME.

MR diffusion tensor imaging also revealed significantly decreased mean fractional anisotropy values in the cerebellum, indicating damage of cerebellar white matter in FAME, compared with controls and patients with essential tremors [70]. The same authors also objectified a global and localized cerebellar gray matter reduction of FAME patients [71]. Afterwards, white matter alterations in other brain regions were extensively disclosed, along with increased white matter volume in the right cerebellum and right sagittal stratum. The authors also found that the cerebellar white matter volume was positively correlated with SEP amplitude (P25–N33), linking cerebellar and cortical hyperexcitability. These findings supported the idea that FAME was a network disorder, involving a wide range of cortical and subcortical structures, including the cerebellum, thalamus, thalamocortical connections, and corticocortical connections (see Section 7) [72]. Functional magnetic resonance imaging (fMRI) studies have also investigated cerebellum–cerebral connectivity and revealed that abnormal functional connectivity between the cerebellum and cerebrum exists in the resting state of patients with FAME [73,74].

## 6. Genetics

### 6.1. Identification of FAME Loci

To date, four loci have been linked to FAME in various pedigrees. Therefore, four variants of FAME have been recognized: FAME1 (8q23.3–q24.1), FAME2 (2p11.1–q12.1), FAME3 (5p15.31–p15.1), and FAME4 (3q26.32–3q28).

FAME1 was mapped to chromosome 8q23.3–q24.1 in Japanese pedigrees and a founder effect was hypothesized [11,12].

FAME2 locus was initially identified on chromosome 2p11.1–q12.2 in an Italian pedigree showing additional features to the core clinical and neurophysiological characteristics of FAME by Guerrini et al. [14]. Three families from Southern Italy sharing the typical clinical and neurophysiological features of Japanese families were subsequently reported with linkage to the same locus as the ADCME pedigree (2p11.1–q12.2) [15,16]. Madia et al. demonstrated that a common founder haplotype was shared by five FAME2 families from Southern Italy [75]. A founder effect was confirmed in a further Italian pedigree [43].

FAME3 locus was identified in a French family, mapping to chromosome 5p15.31–p15.1 [22,23]. FAME3 pedigrees have also been subsequently described in China [24,25,26].

FAME4 was lastly mapped to chromosome 3q26.32–3q28 in a Thai family [27].

Several linkage studies progressively narrowed the FAME1 locus to 4.17 Mbp [76,77] and the FAME2 interval to 9.78 Mbp [28,75,78].

### 6.2. Non-Coding TTTTA and TTTCA Repeat Expansions

Several coding variants were identified within FAME loci in a minority of pedigrees, involving the *SLC30A8* (FAME1) [77], *ADRA2B* (FAME2) [79], and *CTNND2* (FAME3) [80] genes and several FAME genes have been suggested when using WES. However, none of these candidate causative genes had sufficient evidence to guarantee their association with FAME [34].

The finding of clinical anticipation in Japanese pedigrees [34] led to the hypothesis of a mutation not detectable using classical NGS could be responsible for FAME. Therefore, trinucleotide or polynucleotide repeat expansion, located either in the coding or non-coding regions, was suspected [81].

Indeed, in 2018 Ishiura et al. [82] identified the expansion of noncoding TTTCA and TTTTA pentanucleotide repeats in intron 4 of *SAMD12* (which encodes sterile α -motif domain-containing 12) as the causative mutation of FAME1. Moreover, in two families in which repeat expansions in *SAMD12* were excluded, similar expansions of TTTCA and TTTTA repeats were detected in *TNRC6A* located on16p21.1 (which encodes trinucleotide repeat-containing 6A; FAME6) and *RAPGEF2* located on 4q32.1 (which encodes Rap guanine nucleotide exchange factor 2; FAME7). Accordingly, the expansions of the same pentanucleotide repeat motifs were supposed to be involved in the pathogenesis of FAME, rather than the functional alteration of genes in which the expanded repeats were located. In addition, an RNA-mediated toxicity mechanism was hypothesized, since the presence of RNA foci containing UUUCA repeats in nuclei of neuronal cells in autopsied brains was observed [82]. To conclude, the expanded TTTCA repeats were proposed to be primarily causative of FAME1, since expansions of TTTTA repeats were observed in a limited proportion of control subjects, and RNA foci with UUUUA repeats were not detected.

The finding of non-coding TTTCA and TTTTA pentanucleotide repeat expansions in *SAMD12* was confirmed in Chinese pedigrees [37,83,84,85] and in an Indian community [86]. Furthermore, intronic expansions have also been identified in the other FAME linkage intervals, including ATTTT and ATTTC in the first introns of *STARD7* (which encodes StAR-related lipid transfer domain-containing 7; FAME2) [40], TTTTA/TTTCA repeat expansions in the first intron of *MARCH 6* (FAME3) [87], and *YEATS2* (FAME4) [88]. It is noteworthy that the TTTCA insertion and expansion were always accompanied by the TTTTA repeats and confirmed as necessary for the development of FAME. The identified genes did not share common pathways, supporting that the pathological mechanism was more likely related to the type of expansion, regardless of the altered gene function. Indeed, the pentameric expansions were demonstrated to not alter either mRNA or protein levels [87,88]. This observation, along with the localization of the FAME-related expansions within intronic regions, further supported the hypothesis of RNA-mediated toxicity (see Section 7).

While triplet repeat expansions have been reported in various neurological diseases (i.e., spinocerebellar ataxias, DRPLA, Friedreich’s ataxia, Huntington’s disease, fragile X syndrome, and myotonic dystrophy), similar pentameric repeat expansions have been detected only in spinocerebellar ataxia 37 (SCA37), in the first intron of *DAB1* [89,90]. This gene has transcripts specifically expressed in the cerebellum, differently from the FAME genes, which are widely expressed within the CNS.

### 6.3. Genotype-Phenotype Correlation

Clinical anticipation in the onset age of cortical tremors or seizures had been observed in Japanese pedigrees even before identifying TTTTA and TTTCA expansions [5,34,91] and suggested that a repeat expansion mechanism could be involved in the pathogenesis. Similarly, in the first reported European pedigrees, a more severe expression of the neurologic phenotype through successive generations was observed, suggesting anticipation [13].

When TTTTA and TTTCA expansions were identified, Ishiura et al. also showed that the length of the expanded repeats tended to be unstable over consecutive generations and appeared inversely correlated to the age at cortical tremor or seizure onset [82]. This correlation has been further confirmed in other pedigrees and the other FAME genes, supporting that progressive increase in the TTTCA/TTTTA repeat length over successive generations provides a molecular explanation for the anticipation observed in FAME [83,84,85,92,93].

## 7. Pathophysiology

Along with features of cortical hyperexcitability, cerebellar clinical and neuroradiological signs have been reported in FAME pedigrees [22,47,48]. Moreover, several imaging studies indicate cerebellar alterations and abnormal functional connectivity between the cerebellum and cerebrum (see Section 5).

Cerebellar dysfunction in FAME has been further demonstrated in post-mortem histological studies in a Dutch pedigree, which disclosed marked loss and morphological changes (i.e., somal sprouting and loss of the dendritic tree) of Purkinje cells in the cerebellar cortex with no or limited changes in the sensorimotor cortex [37,47,48]. In a single South African pedigree, neuropathology investigations also showed Purkinje cell loss, dentate atrophy, and neuronal cell loss, with gliosis in the olives and pallidum [94]. Similar cerebellar features were described in spinocerebellar ataxia subtype 6 (SCA6), caused by *CACNA1A* mutations. These observations indicated a central role of the cerebellum in the pathogenesis of FAME.

Recently, Latorre et al. suggested that cortical tremor is part of a spectrum of cortical myoclonus, which is due to the evolution from a spatially limited focus of excitability to the recruitment of more complex networks capable of sustaining repetitive activity. They also observed that cortical tremor distinguishes itself from the other forms of cortical myoclonus by its rhythmicity and proposed that it could be driven by abnormal activity in the cerebellar–thalamic–cortical loop [42].

Dubbioso et al. demonstrated the involvement of subcortical networks (i.e., dorsal column nuclei and the thalamus) using combined TMS and SEPs investigations [66]. Indeed, both cortical tremor and epilepsy have been hypothesized to be the result of decreased sensorimotor cortical inhibition through dysfunction of the cerebellar–thalamic–cortical loop, secondary to primary cerebellar pathology [48,95,96].

Alternatively, a common mechanism could underlie both cortical hyperexcitability and cerebellar degeneration as calcium signaling dysfunction. The presence of additional features, including migraine and night blindness, supported that FAME could be a channelopathy. However, mutations in genes encoding ion channels were excluded and cortical myoclonus is not a feature of SCA6, another repeat expansion disorder, despite the loss of Purkinje cells.

The exact pathogenic mechanisms involving the cerebellum have yet to be elucidated. The detection of RNA foci containing repeats in the nucleus of cortical neurons and Purkinje cells supported an RNA-mediated toxicity mechanism [40,82]. A similar RNA toxic gain-of-function has been evidenced in other non-coding repeat expansion neurodegenerative diseases (e.g., *C9ORF72*-linked ALS/frontotemporal dementia, myotonic dystrophy, fragile X tremor/ataxia syndrome, some spinocerebellar ataxias, and Huntington’s disease-like 2), and further, underlying non-mutually exclusive mechanisms have been hypothesized. They include the functional alteration of RNA-binding proteins sequestrated into the RNA foci, the translation of the expanded transcripts into neurotoxic peptides (namely, RAN proteins, since the translation occurs in a “repeat-associated non-AUG” way), and the loss or the overexpression of linked genes [97,98,99,100]. Nevertheless, there is no evidence of RNA-mediated toxicity in FAME, since no RAN neuropeptides nor RNA-binding proteins sequestrated into the RNA foci have been demonstrated.

## 8. Treatment Strategies and Future Perspectives

The medical treatment of FAME patients is based on drugs with both anti-seizure and anti-myoclonic effects. Commonly, valproate and levetiracetam are first-line medications, associated with benzodiazepines. Among these latter, clonazepam is considered the most effective in FAME, owing to its longer half-life compared to other BDZs. Perampanel might be useful as the first add-on to first-line ASMs and has been demonstrated to be specifically efficacious on refractory cortical myoclonus, even at low doses [64,65,101]. Additionally, piracetam (PIR) is among the first ASMs recognized as efficacious on cortical myoclonus, albeit it is no longer used [102].

Noteworthy, sodium channel blockers such as carbamazepine and gabapentin, which are commonly used in essential tremor, may worsen myoclonus, or lead to status epilepticus in FAME [103,104]. In addition to medical treatment, alternative symptomatic approaches might be considered in FAME. Repetitive TMS has shown therapeutic potential in myoclonus of different etiologies. Specifically, low-frequency (LF) rTMS (stimulus rates of 1 Hz or less) over the primary motor cortex produces an inhibition of motor cortical excitability and has been demonstrated to improve cortical myoclonus in one individual with myoclonus differently from FAME [105,106]. Moreover, the advances in understanding the pathogenesis of FAME, and particularly the detection of RNA foci containing repeats in neuronal cells, pave the way to novel therapeutic possibilities, as RNA-targeting treatments (i.e., antisense oligonucleotides, RNA-targeting Cas9) [107]. Indeed, antisense oligonucleotides (ASOs) capable to bind the RNA pentamers expanded and may prevent the formation of RNA aggregates or the translation of the pentameric repeats into neurotoxic peptides [108]. ASOs have been proposed as therapeutic strategies for diverse neurodegenerative repeat expansion disorders, including Huntington’s disease. Additionally, ASOs-based treatment has been recently introduced for spinal muscular atrophy [109].

As FAME represents a novel non-coding repeat expansion disease model [89,90], the elucidation of the underlying pathomechanisms is expected to provide precision therapeutic strategies, possibly applicable to further neurodegenerative non-coding disorders.

## Figures and Tables

**Figure 1 cells-12-01617-f001:**
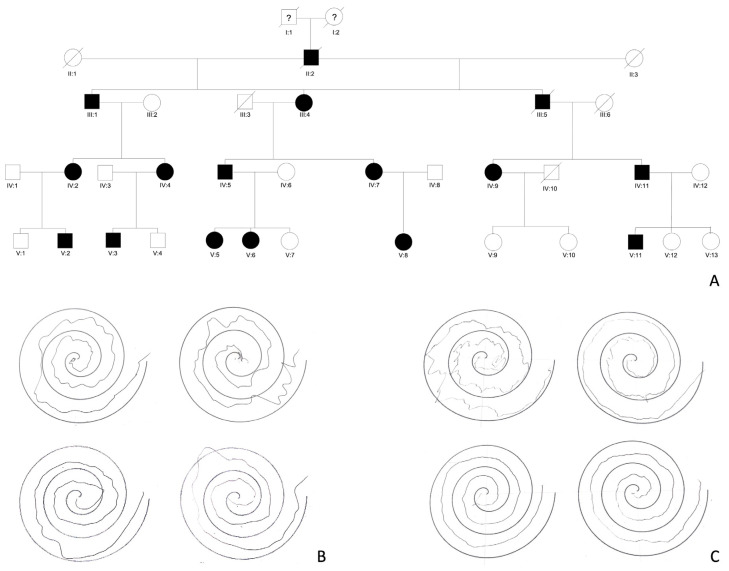
(**A**) A South Italian pedigree affected with FAME. The members with “?” have unknown medical history. This family has been partially described in 2019 as family 11 [40] and in 2003 as Family B [15]. (**B**) Hand spirals drawn by individual IV:2 at 61 years old upward; individual V:2 at 34 years old downward. (**C**) Hand spirals drawn by individual IV:4 at 54 years old upward; individual V:3 at 31 years old downward.

**Figure 2 cells-12-01617-f002:**
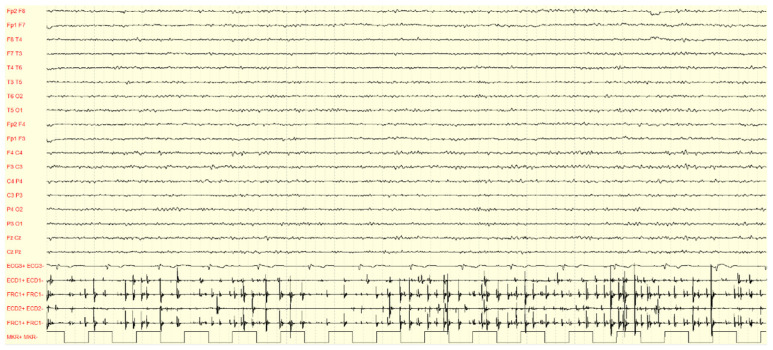
EEG-EMG polygraphy of individual IV:2 recorded at the age of 61 years old. The image shows a normal background of EEG activity and spontaneous brief myoclonic jerks recorded over the EMG channels (ECD1: left extensor communis digitorum; FRC1: left flexor carpi radialis; ECD2: right extensor communis digitorum; FRC2: right flexor carpi radialis).

## Data Availability

Not applicable.

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
