# Peer review of "Familial Adult Myoclonus Epilepsy: A Non-Coding Repeat Expansion Disorder of Cerebellar–Thalamic–Cortical Loop"

_cells, 2023, doi:10.3390/cells12121617_

Round 1
Reviewer 1 Report
Comprehensive and nicely written review. No changes in the content are necessary from my perspective.
Author Response
We thank the Reviewer for his/her kind comment.
Reviewer 2 Report
The paper looks concise and clearly written. It will provide useful information on the current knowledge of the genetically heterogeneous disorder, FAME, caused by similar non-coding pentanucleotide repeat expansions.
Author Response

(The authors gave the same response as above.)

Reviewer 3 Report
This is a well written review touching upon history, clinical, neurophysiological and molecular pathogenesis of familial adult myoclonus epilepsy. Authors have delt with a nosological issue on this entity.
I have one small suggestion: please provide more detailed description on RNA-gain of mechanism written in 9-10. For example, if RNA foci was demonstrated by multiple labs, I think it would enhance its importance. Another suggestion would be to describe RNA binding protein in FME. If there is not enough publication, please briefly write so.
Author Response
We thank the reviewer for the positive comment and the valuable suggestion. We apologize for not having elaborated on this aspect. In paragraph 7.2 “Non-Coding TTTTA and TTTCA Repeat Expansions” (lines 407-411) we mention that an RNA-mediated toxicity mechanism has been hypothesized since the identification of intronic TTTCA and TTTTA in the SAMD12 gene by Ishihura et al. The authors indeed observed the presence of RNA foci containing UUUCA repeats in nuclei of neuronal cells in autopsied brains.
In paragraph 8 “Pathophysiology” (lines 478-482) we aim to deepen the pathogenetic mechanisms of FAME, both neurophysiological and molecular ones. Therefore, as you suggested, we have completed the last portion of this paragraph as follows: “The detection of RNA foci containing repeats in the nucleus of cortical neurons and Purkinje cells supported an RNA-mediated toxicity mechanism [82, 86]. A similar RNA toxic gain-of-function has been evidenced in other non-coding repeat expansion neurodegenerative diseases (e.g. C9ORF72-linked ALS/frontotemporal dementia, myotonic dystrophy, fragile X tremor/ataxia syndrome, some spino-cerebellar ataxias, and Huntington’s disease-like 2) and further underlying non-mutually exclusive mechanisms have been hypothesized. They include the functional alteration of RNA-binding proteins sequestrated into the RNA foci, the translation of the expanded transcripts into neurotoxic peptides (namely RAN proteins, since the translation occurs in a “repeat-associated non-AUG” way) and the loss or the overexpression of linked genes [99, 100, 101, 102]. Nevertheless, there is no evidence about RNA-mediated toxicity in FAME, since no RAN neuropeptides nor RNA-binding proteins sequestrated into the RNA foci have been demonstrated”.